# Integrated Biorefinery and Life Cycle Assessment of Cassava Processing Residue–From Production to Sustainable Evaluation

**DOI:** 10.3390/plants11243577

**Published:** 2022-12-18

**Authors:** Larissa Renata Santos Andrade, Raul José Alves Felisardo, Ianny Andrade Cruz, Muhammad Bilal, Hafiz M. N. Iqbal, Sikandar I. Mulla, Ram Naresh Bharagava, Ranyere Lucena de Souza, Lucas Carvalho Basilio Azevedo, Luiz Fernando Romanholo Ferreira

**Affiliations:** 1Graduate Program in Process Engineering, Tiradentes University, Av. Murilo Dantas, 300, Farolândia, Aracaju 49032-490, SE, Brazil; 2Biomass Technology Laboratory, Université de Sherbrooke, 2500 Boul, de L’Université, Sherbrooke, QC J1K 2R1, Canada; 3Institute of Chemical Technology and Engineering, Faculty of Chemical Technology, Poznan University of Technology, Berdychowo 4, PL-60965 Poznan, Poland; 4Tecnologico de Monterrey, School of Engineering and Sciences, Campus Monterrey, Ave. Eugenio Garza Sada 2501, Monterrey 64849, Mexico; 5Department of Biochemistry, School of Allied Health Sciences, REVA University, Bangalore 560064, India; 6Laboratory for Bioremediation and Metagenomics Research (LBMR), Department of Environmental Microbiology (DEM), Babasaheb Bhimrao Ambedkar University (A Central University), Vidya Vihar, Raebareli Road, Lucknow 226025, India; 7Institute of Technology and Research, Av. Murilo Dantas, 300, Farolândia, Aracaju 49032-490, SE, Brazil; 8Institute of Agricultural Sciences, Federal University of Uberlândia, Campus Glória, Uberlândia 38410-337, MG, Brazil

**Keywords:** circular economy, bioconversion, biofuels, bioproducts, environmental impacts

## Abstract

Commonly known as a subsistence culture, cassava came to be considered a commodity and key to adding value. However, this tuber’s processing for starch and flour production is responsible for generating a large amount of waste that causes serious environmental problems. This biomass of varied biochemical composition has excellent potential for producing fuels (biogas, bioethanol, butanol, biohydrogen) and non-energetic products (succinic acid, glucose syrup, lactic acid) via biorefinery. However, there are environmental challenges, leading to uncertainties related to the sustainability of biorefineries. Thus, the provision of information generated in life cycle assessment (LCA) can help reduce bottlenecks found in the productive stages, making production more competitive. Within that, this review concentrates information on the production of value-added products, the environmental impact generated, and the sustainability of biorefineries.

## 1. Introduction

The increasing generation of waste, the depletion of fossil fuel reservoirs, and the consequent fluctuation in fuel prices have forced nations to develop policies to encourage the use of renewable energy [1]. Thus, producing biofuels and non-energy products from biomass has aroused interest due to advantages such as reduction of greenhouse gas emissions and the possibility of generating new jobs [2]. Moreover, the linear economy model (extraction–production–use) has been gradually replaced by the concept of circular economy, which aims to maximize resource efficiency and waste reduction [3]. In this sense, implementing biorefineries is an indispensable strategy for executing the circular economy plan, as it will stimulate many economic opportunities, in addition to the sustainable transformation of waste into raw materials [4].

Cultivated mainly in the tropical region and in some of the world’s poorest regions, cassava production doubled in just over two decades [5]. Thus, the residues generated in cassava processing stand out as an important source of resources for biorefineries. These residues are generated during the production of flour and starch and differ in their composition. In general, peels from the initial processing, fibrous by-products generated during milling and sieving, and starch residues are generated after decanting starch and wastewater effluents. These residues, when discarded without treatment, cause contamination of soil and water bodies [6]. On the other hand, cassava residue-based biorefineries can encourage industrial expansions of the tuber and the supply of sustainable energy for starch processing [7]. However, it is of great importance that waste conversion into energy and non-energy products is carried out with minimal external resources, such as water, fuel, electricity, and land use [8].

Even if there are some advances in the prospective study of biorefineries of cassava residues, it is still essential to evaluate the entire system’s sustainability. In that sense, the life cycle assessment (LCA) is a systematic methodology (ISO 14000) widely used to identify the environmental effects generated by processes or products [4]. Therefore, this review gathers information on the recovery of these residues (peels, roots, bagasse, and wastewater), aiming to encourage the economic-sustainable recovery of these materials within the biorefinery concept. The published literature on the evaluation of cassava processing residue biorefinery from 2012 to 2022 was examined. Initially, this article highlights the generation of energy and non-energy bioproducts from the various cassava residues. Then the existing challenges of life cycle assessment studies and future research directions are recommended accordingly.

## 2. Biorefinery: Mechanisms and Development

Biorefineries are facilities that aim to generate energy and products from animal, plant, or microorganism-origin organic materials. The feasibility and development of a biorefinery are related to the optimization of variables and processes from identifying critical factors such as types of raw materials and their supply chains, technical and economic evaluation, and identification of future trends [9].

According to Liu et al. [10], biorefinery development is divided into four generations. The first generation was marked by limited raw materials, technology, and products. The second generation had a more varied production but was still limited in input and technology. The third-generation diversified inputs, technologies, and production, encouraging productive efficiency and reducing environmental impacts during the process. Building on the previous generation, the fourth generation focused on reducing the economic cost and increasing the ecological benefit.

The concept of biorefinery emerged as an alternative for reducing pollution and valuing biomass [11]. This system uses biomass as a raw material in different processes to generate a massive variety of biofuels and bioproducts. This input can be classified as virgin material (plants of aquatic or terrestrial habitats that occur naturally) or residual material (from municipal, agricultural, and industrial activities) [12].

The leading technologies for converting these raw materials are divided into thermochemical, physical, chemical, and biological processes (Figure 1). Thermochemical processes use high temperatures to convert waste into fuels, electricity, heat, and value-added products. Physical processes include mechanical treatment such as physical phase separation and particle size reduction. Chemical processes use chemical agents to convert feedstock into liquid biofuels and bioproducts. Finally, biochemical processes use biological agents to convert raw materials into liquid and gaseous biofuels and bioproducts.

## 3. Cassava Waste Used in Biorefineries: Supply Chain

Cassava (*Manihot esculenta Crantz*) is part of the food culture of many countries, being used as raw material for various products. It is a non-grain crop in which the root is rich in starch and straw in cellulose [13]. Due to the production process of this food, of which starch and flour stand out, large amounts of waste are generated. According to Veiga et al. [14], the residues of the cassava industry include stalk from the harvest (about 63% of the cassava root mass), bagasse (scraped pulp and peel), and large volumes of effluent generated in starch production. Most cassava splints are discarded in the field, left as waste [8], while a small amount (10–20%) is used in planting [15].

Obtaining cassava starch consumes a lot of water and generates a large flow of wastewater. On the other hand, the peeling and washing of the raw material in traditional installations generally does not use automated processes, most of which are produced during the peeled roots’ pressing. This process has a smaller amount of effluent but is more concentrated. This effluent, without any treatment, is released near the producing region, polluting the soil and rivers. In addition, the high content of organic compounds present in the effluent generates a considerable fall in the oxygen levels of rivers [8].

Thus, the availability of raw material in cassava residue biorefineries is directly linked to the processing of this tuber. The needs of these industrial plants range from storing the raw material (cassava residues) to the characterization of the final product. This way, small or large-scale processors can support their supply chain. An additional route can also be entered when incentives are given to subsistence producers [16].

Thus, the circular economy proposes a configuration of industrial activity to focus on reducing pollution and converting waste into value-added products [8]. This concern with the balance of resources within the industrial sector led to the development of the incorporation of environmental policies in the different stages of the production chain. In a green supply chain, parameters that affect socio-environmental and economic aspects are analyzed to ensure quality products with reduction of the carbon footprint [17].

### 3.1. Energy and Non-Energy Bioproducts

Given its complex biochemical composition, high starch content, and considerable amount, the disposal of cassava waste becomes a significant environmental problem. An alternative that has received attention is the use of these materials in biorefineries since their high organic content gives them an excellent potential for bioconversion into value-added products [6,18]. Therefore, Table 1 highlights the main bioproducts generated from different cassava residues.

#### 3.1.1. Biogas

Biogas has been touted as a promising source of clean energy. Among other ways, it can be obtained from the digestion of agricultural residues, such as cassava waste [40]. Generally, biogas is generated in an anaerobic digester where the substrate is converted into energy [40,41]. This digester is also known as a biogas digester [42], offering a renewable energy source as methane can be used to generate heat and/or vehicular fuel [40].

Several works in the literature have reported cassava residues’ use in biogas production. Lin et al. [40] evaluated the life cycle of a biogas system for cassava processing in Brazil to close the loop in the water–waste–energy–food nexus. Achi et al. [32] studied the improvement of biogas production from cassava wastewater employing manure co-digestion, and zeolite and biochar additives. Sawyerr et al. [42] designed a domestic biogas digester using co-digested cassava, vegetable, and fruit waste. Cremonez et al. [43] evaluated the process of biphasic anaerobic digestion of a polymer based on cassava starch, determining the ideal organic load to obtain the best results in terms of solids removal and methane and hydrogen production. Jiraprasertwong et al. [30] investigated biogas production from cassava wastewater using an up-flow three-stage anaerobic sludge blanket reactor.

#### 3.1.2. Bioethanol

Bioethanol is a renewable energy source used as a liquid vehicle fuel [44] and can partially or replace gasoline use [13]. The use of cassava residues for bioethanol production is due, in general, to the high content of starch [22]. The production of this fuel has received attention due to the potential of green technology to alternatively use the burning of biomass and fossil fuels, thus reducing greenhouse gas emissions [45].

Murata et al. [22] investigated the potential of cassava pulp and peel as substrates for ethanol production. Amalia et al. [45] analyzed the bioconversion and valorization of industrial residues based on cassava into bioethanol gel and its potential application as a clean fuel for cooking. Zhang et al. [46] dedicated their studies to develop an efficient technology for the bioconversion of cassava into bioethanol without high-temperature digestion. In this work, a non-digestive strategy via mechanical activation and pre-treatment with metallic salts was used. Rewlay-ngoen et al. [19] evaluated the environmental performance of ethanol produced from cassava pulp and its use as a transport fuel.

#### 3.1.3. Butanol

Butanol is an alcohol that is industrially used as a solvent [47]. It has been considered an alternative vehicle fuel due to specific gasoline-like characteristics such as energy density, vaporization heat, and non-corrosiveness [25,48]. Due to this, there is great interest in the production of butanol through the fermentation of biomass, such as cassava residues [24].

Saekhow et al. [25] studied the enzymatic hydrolysis of cassava stems in the production of butanol. Li et al. [49] investigated the improvement in butanol production in *Clostridium acetobutylicum* SE25 through phase shift acceleration by regulating the pH of different phases of cassava flour. Li et al. [50] used the direct fermentation process of cassava flour to produce butanol. Lu et al. [23] used a cassava bagasse hydrolyzate in a fibrous bed bioreactor with continuous gas stripping to produce butanol from batch fermentation.

#### 3.1.4. Lactic Acid

Lactic acid is a monomer with wide application in the chemical, pharmaceutical, agricultural, textile and plastic industries [51]. It is an important chemical product of commercial interest that can be obtained microbially in the fermentation process [52,53]. Therefore, the recovery of cassava waste for the production of lactic acid is of great importance. Its most relevant applications involve the synthesis and processing of poly-acid, as well as biodegradable and biocompatible plastics [53,54].

Chen et al. [29] used cassava bagasse to produce lactic acid based on simultaneous saccharification and co-fermentation. By these same processes, Chookietwattana [55] produced lactic acid from cassava starch by *Lactobacillus plantarum* MSUL 903. By using amylolytic *Lactobacillus plantarum* MSUL 702, Tosungnoen et al. [28] investigated the generation of lactic acid from wastewater containing cassava starch.

#### 3.1.5. Succinic Acid

Succinic acid, also called butanedioic acid, is an organic acid that is the final product of the anaerobic fermentation of some microorganisms. It has received a lot of attention in biorefineries because it is a chemical product produced from lignocellulosic materials that can be used in the synthesis of many compounds [56,57], such as butanediol, butyrolactone, tetrahydrofuran, fatty acids, and biodegradable polymers [58].

The synthesis of this acid from cassava residues as lignocellulosic material has become the subject of some studies. Shi et al. [56] investigated the process of economically improved fermentation of succinic acid from cassava bagasse hydrolyzate using *Corynebacterium glutamicum* immobilized on porous polyurethane filler. Thuy et al. [27] used cassava root as a substrate for succinic acid fermentation using *Actinobacillus succinogenes* ATCC55618. Sawisit et al. [26] investigated production from cassava pulp using genetically modified *Escherichia coli* KJ122.

#### 3.1.6. Biohydrogen

Biohydrogen can be produced from the dissociation of the water molecule (thermolysis, electrolysis, and photolysis), thermochemical conversion (pyrolysis, gasification, combustion, liquefied), and biological conversion of biomass (biophospholysis, dark fermentation, photofermentation) [35,59]. The production of green hydrogen from cassava residues is a promising alternative and much discussed among authors. Martinez-Burgos et al. [35] evaluated biohydrogen production using cassava wastewater and two microbial consortia from different environments. Lin, Cheng, and Murphy [38] investigated the biohydrogen yield from the co-fermentation of mixed cassava residue and pig manure submitted to microwave-assisted hydrothermal pretreatment.

On the other hand, Meier et al. [34] investigated the optimization of biohydrogen production from anaerobic biodigestion cassava wastewater and swine wastewater with glycerol additions. Pason et al. [39] studied sustainable hydrogen production using untreated cassava pulp and a consortium of thermophilic anaerobic bacteria. Mari et al. [36] evaluated the energy potential of cassava starch wastewater in a two-stage system (BioH_2_ + BioCH_4_) composed of anaerobic sequencing batch biofilm reactors (AnSBBRs). Hydrogen production using cassava starch wastewater was also evaluated by Corbari et al. [37], but using a fixed-bed anaerobic reactor (UAFBR).

## 4. Sustainable Production and Environmental Impact

Cassava ranks second in the world production position of the root and tuber family. Previously, it was considered a subsistence crop in tropical and subtropical zones. Now, it is seen as a commodity and key to adding value [5]. In addition, many countries dependent on petroleum-derived fuels are among the largest producers of this biomass [16].

Waste from cassava processing is rich in organic matter and suspended solids, and its improper disposal can cause negative impacts on the environment. Therefore, the concept of a biorefinery and the feasibility of converting these residues into biofuels and different bioproducts presents the possibility of advancement, as the countries with the highest production of cassava are also some of the countries with the greatest lack of energy production [6]. On the other hand, there is a foreseeable limitation in the implementation of cassava waste biorefineries due to uncertainties related to the technical–economic and environmental feasibility of the different biorefinery routes [60].

In this sense, the provision of details on the environmental impact generated in life cycle assessment (LCA) can help reduce the obstacles encountered, making production more competitive with products of fossil origin [1]. LCA is a standard and widely accepted technique that evaluates the impacts generated by a product or service, from the raw material’s acquisition to its reuse or disposal. This environmental management tool considers the cost and environmental and social assessments of the life cycle. Thus, LCA can be used in the planning of the biomass supply chain to evaluate the environmental impacts generated in the areas of supply, pre-processing, transportation to biorefineries, production, and distribution [2]. Figure 2 highlights the main steps for implementing the life cycle assessment in a biorefinery. However, Kosamia et al. [61] highlight the importance of applying these methodologies after the success of laboratory-scale experiments and before the start of industrial-level processes.

### 4.1. Life Cycle Environmental Performance

Several environmental impacts associated with waste biorefineries can be analyzed through the LCA. In general, eight categories of environmental impact can be found in the literature: abiotic destruction potential, acidification potential, eutrophication potential, ecotoxicity potential, global warming potential, ozone bed destruction potential, human toxicity potential, and photochemical oxidation potential [10]. However, this review aims to evaluate the works directed to LCA in cassava residue plants, as observed in Table 2. For this section, information was extracted on the following categories most discussed in the eleven selected articles: global warming potential (GWP), acidification potential (AP), human toxicity potential (HTP), and photochemical ozone generation potential (POGP).

#### 4.1.1. Global Warming Potential (GWP)

Global warming potential (GWP) refers to the ratio of the warming caused by a substance such as greenhouse gases (CO_2_, CH_4_, and N_2_O) to the warming caused by a similar mass of CO_2_. This is one of the most used impact categories in the environmental assessment of bioenergy. The results of this indicator have been discussed based on scenarios with different processes, raw materials, and energy supply modes.

With a view to evaluating the environmental impact of cassava residues on energy production, Lyu et al. [13] compared ethanol production systems from the root, straw, and whole cassava. The authors considered the use of the whole cassava plant as the most economical and environmentally correct raw material for the production of bioethanol. However, a greater use of straw in integrated processes can help to reduce the effects of GWP. In general, the authors highlighted the integration of processes, mainly in the planting phase, and the reduction of the input of fertilizers as promising approaches for the reduction of GWP.

In their work, Lansche et al. [66] also scored agricultural practices as contributors to GWP. They showed that the biggest contribution to this category is related to CO_2_ and N_2_O emissions, with 60% and 36%, respectively. Nitrogen fertilization has been shown to be the source of direct N_2_O emissions, while exhaust gases from tractors and machines used in the modern cultivation system are the main source of CO_2_ emissions compared to traditional production.

On the other hand, Rewlay-ngoen et al. [19] estimated GHG emissions during the production of ethanol from cassava pulp and during its application as transport fuel. The authors indicated electricity consumption during ethanol conversion and starch processing, and N_2_O production during nitrogen fertilizer use, as the main contributors to climate change emissions. When evaluating the benefits of applying biogas technology in cassava starch factories, Hansupalak et al. [67] also highlighted N_2_O emissions during the use of fertilizers in the agricultural phase as the main contributor to the carbon footprint. On the other hand, in the scenario without biogas production, the contributions of CH_4_ emissions from wastewater, fossil CO_2_ emissions during the combustion of fuel oil, and greater use of electricity in the starch production phase were highlighted.

#### 4.1.2. Acidification Potential (AP)

The acidification potential (AP) is also widely used in the assessment of the environmental impact of biofuels and vehicles, as it provides information related to their influence on the soil. In this perspective, Padi, Chimphango, and Roskilly [65] compared AP rates in cassava waste bioenergy facilities with the current fossil energy scenario. The authors pointed out that the environmental management of starch waste implementation could improve PA by approximately 95%. The recovery of these residues through the production of biogas and biofertilizers inhibits emissions associated with the soil in cases of wastewater disposal and burning of stalks in the vicinity of the facilities.

Aiming at the application of biogas from cassava wastewater as transport fuel, Papong et al. [68] evaluated the PA in all stages of production (anaerobic digestion, cleaning, and upgrading, biomethane compression, compressed biomethane transportation, gas station, combustion). The upgrading stage, biogas production, and biomethane compression showed the highest contributions among all technologies due to higher electricity consumption. In another study, Lansche et al. [66] suggest replacing electricity and conventional heat with biogas from cassava processing residues during the production of starch and chips. Compared to the entire production chain, the results show that the greatest PA impact is related to direct SO_2_ and N_2_O emissions in the agricultural cultivation phase.

With regard to bioethanol production, Lyu et al. [13] examined simulations of integrated processes in the conversion of this biofuel from different parts of cassava. They pointed out that the use of the entire plant as raw material would reduce land use and, consequently, the contamination caused by excess chemical fertilizers in the soil. In contrast, the direct use of straw in the production of bioethanol contributed to the higher consumption of fertilizers and the increase in AP rates (8.73 kg SO_2_ eq.), when compared to cassava root (4.35 kg SO_2_ eq.) and to the entire plant (4.04 kg SO_2_ eq.). In the work by Zhan et al. [62], they use a model that can be used in the production of ethanol from different feedstocks. The application of this model in other studies contributes to a more detailed investigation of the impacts caused in different bioethanol plants, in addition to providing the scientific basis for future evaluations.

#### 4.1.3. Human Toxicity Potential (HTP) and Photochemical Ozone Generation Potential (POGP)

In addition to the categories already mentioned, Lyu et al. [10] highlighted the environmental impacts characterized in relation to HTP and POGP. Two different scenarios were analyzed, considering only cassava straw (CS) and whole-plant cassava (WPC), the latter being studied in two ways: WPC-1, in which the available fermentable sugar was sent to the distillation unit, and WPC-2, in which the cellulosic sugar was sent to the liquefaction unit. The uncertainty analysis of the environmental impact of the bioethanol life cycle showed that when using WPC-2 there are greater benefits than producing bioethanol with CS, and the HTP reduction is more than 260 kg DCB eq. in the production of 1000 L of bioethanol. Regarding the photochemical potential of ozone generation, the authors noticed that the amount of ozone generated considering CS was 0.550 kg Ethene eq., while with WPC-2 it was 0.403 kg Ethene eq.

In studies by Rewlay-ngoen et al. [19], the environmental performance of ethanol produced from cassava pulp was investigated under four scenarios: (S1) ethanol plant used fuel oil in a boiler to produce steam; (S2) considered the same conditions as in Scenario 1, but the production of cassava pulp was based on economic allocation; (S3) ethanol plant used biogas to replace fuel oil in a boiler; and (S4), which considered the same conditions as in Scenario 3, but the cassava pulp was based on economic allocation. In general, the results in terms of HTP were: S1 = 8.34 kg 1,4-DB eq., S2 = 6.62 kg 1,4-DB eq., S3 = 8.20 kg 1,4-DB eq., S4 = 6.48 kg 1,4-DB eq. Thus, the best scenario for HTP is S4, the ethanol plant that uses biogas to replace fuel oil for steam production and in which the impacts of cassava pulp are allocated based on economic value.

Zhan et al. [62] performed the life cycle assessment of the optimized cassava ethanol production process based on operational data from the Guangxi plant in China. The objective of this work was to point out a more economical and environmentally correct production route throughout the production cycle, considering planting, harvesting, preparation, and transport, even in the conversion of bioethanol. Different cases were considered: “Base case”, in which a simulation was made based on known factory data, considering the traditional production process of clinker fermentation (CF), the alcohol concentration of the fermentation broth being 15% *v*/*v*; “Case a”, traditional production process of clinker fermentation and alcohol concentration of the fermentation broth of 12.5% *v*/*v*; “Case b”, traditional production process of clinker fermentation and alcohol concentration of the fermentation broth of 17.5% *v*/*v*; “Case c”, traditional production process of clinker fermentation and alcohol concentration of the fermentation broth of 20% *v*/*v*; and “Case d”, with traditional production process of fermentation of the raw material and alcohol concentration of the broth, fermentation is 15% *v*/*v*. The results showed that for these two environmental categories, “Case c” is the one that contributes the least to the impacts, and on the other hand, “Case a” is the one that contributes the most. Regarding the “Base case”, the impact on POGP is 0.088457 kg C_2_H_4_ eq., while the contribution to HTP is 174.59 kg 1,4-DB eq. This contribution to HTP is reduced by 0.13 kg 1.4-DB eq. when “Case d” is considered. Thus, it can be seen that the increase in the alcohol concentration of the fermentation broth is beneficial for the HTP and POGP categories.

## 5. Methodology

This review summarizes the most recent research on cassava processing residue biorefineries, considering the types of residues and optimal predictive methodologies for developing these biorefineries. The articles were retrieved using the keywords: “Cassava waste” and “biorefinery” from the available ScienceDirect and Scopus databases. The scope covers articles on biorefineries and cassava waste recovery and focuses on peer-reviewed research articles published between the years 2012 and 2022. As a result, 38 papers were selected, 16 articles from ScienceDirect and 22 from Scopus. Figure 3 presents the survey of these articles by year of publication. This figure shows a reduction in the number of publications on the topic “Biorefineries of cassava processing waste” from 2017 onwards. On the other hand, from 2020 onwards, there has been a considerable increase in publications highlighting the topic’s relevance. This increase may have been driven by strong pressure from global decarbonization, rising fossil fuel prices, and increased waste generation during the COVID-19 pandemic.

## 6. Final Considerations and Future Perspectives

Bioconversion is a circular economy strategy used to reuse large quantities of industrial waste. In this context, biorefineries based on agricultural waste, such as cassava residues, present themselves as a solution for the advancement of pollution. Thus, most often, unrestrictedly discarded cassava residues start to have value when used as raw material in the production of fuels and different bioproducts. However, despite the importance of investments in research focused on the bioconversion of cassava residues into value-added products, the number of studies evaluating the raw material’s productive potential and the environmental impact of the supply chain are still considerably limited.

In addition, it was observed that the works found focused on assessing the environmental impacts of biofuel production (ethanol and biogas) to the detriment of non-fuels. However, the variety of the functional unit and the processes involved makes it difficult to standardize an efficient method for the development of these biorefineries. Another limiting factor is the varied data source used in these works. Although all sources adopt the LCA structure defined by the ISO standard, the database is not unified, hampering the comparison of the results found. The exclusion of information on the environmental burden associated with infrastructure and the limited availability of studies based on the “cradle to grave” perspective”, which take into account the environmental impacts of the product’s end of life, also contribute to less realistic quantifications and, consequently, limitations in future applications.

In fact, many industrial processes have the potential for generating impacts. With regard to biorefineries, regarding the use of biological products, these impacts can be minimized. Life cycle analysis is of great importance in the feasibility study. It was seen that among the characterized impacts, the categories global warming potential, acidification potential, human toxicity potential, and photochemical ozone generation potential are the most studied, and in cassava processing, they deserve a lot of attention.

## Figures and Tables

**Figure 1 plants-11-03577-f001:**
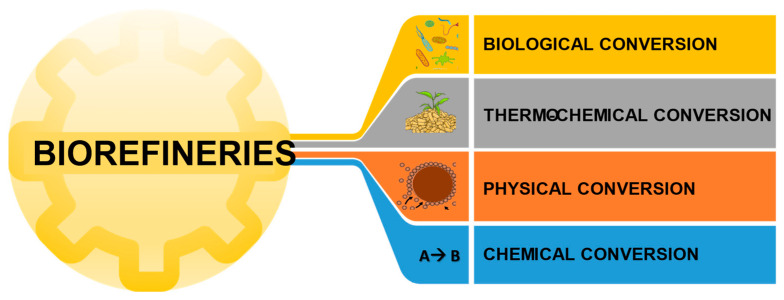
Types of waste conversion processes.

**Figure 2 plants-11-03577-f002:**
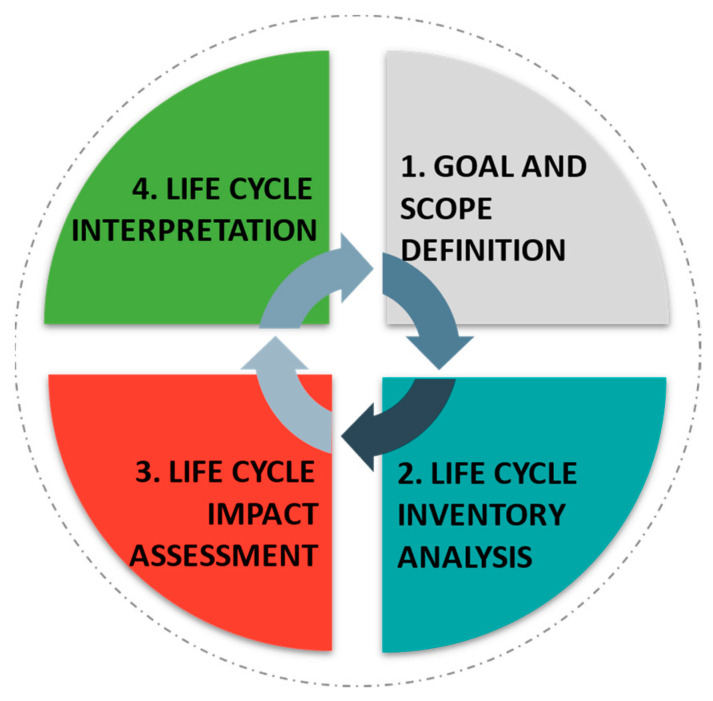
Steps for implementing life cycle assessment.

**Figure 3 plants-11-03577-f003:**
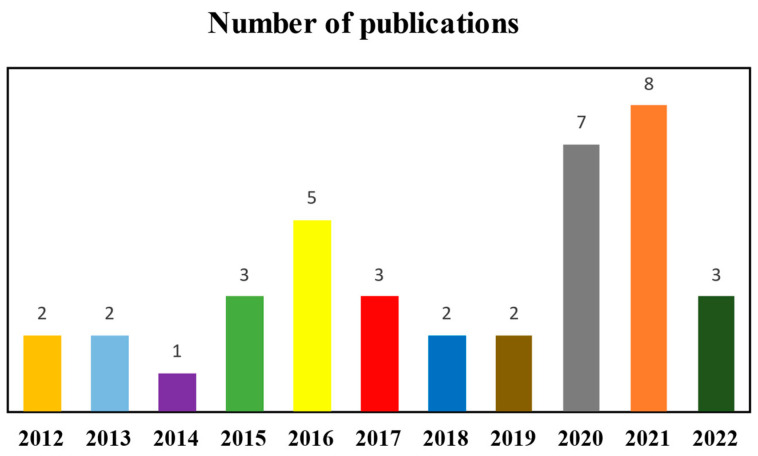
Annual evolution of articles published in the available ScienceDirect and Scopus databases, using the keywords: “cassava waste” and “biorefinery”.

**Table 1 plants-11-03577-t001:** Types of bioproducts generated from cassava residues reported in the literature.

Cassava Waste	Products	References
Pulp	Bioethanol	[19]
Peel	Bioethanol	[16]
Stems	Bioethanol	[20,21]
Pulp and Peel	Bioethanol	[22]
Bagasse	Butanol	[23,24]
Stems	Butanol	[25]
Wastewater, bagasse and Stems	Succinic acid, glucose syrup, bioethanol	[7]
Pulp	Succinic acid	[26]
Root	Succinic acid	[27]
Wastewater	Lactic acid	[28]
Bagasse	Lactic acid	[29]
Wastewater	Biogas	[30,31,32,33]
Wastewater	Biohydrogen	[34,35,36,37]
Mixed waste	Biohydrogen	[38]
Pulp	Biohydrogen	[39]

**Table 2 plants-11-03577-t002:** A summary of key elements of the LCA from articles selected for this review.

Cassava Waste	Products	Functional Unit (FU)	Country	Environmental Analysis (Categories)	References
Whole-plant cassava	Bioethanol	1 t (99.7%, 26,840 MJ) of bioethanol	China	GWP, OLDP, HTP, POGP, AP, EP	[62]
Pulp	Bioethanol	1 km driven by a vehicle	Thailand	CG, AP, FEP, HTP, FD, FOF	[19]
Straw	Bioethanol	1000 L 99.7 vol% bioethanol with 21,200 MJ	China	GWP, AP, EP, HTP, POGP	[44]
Whole-plant cassava, root, straw	Bioethanol	1000 L 99.7 vol% bioethanol with 21,200 MJ	China	GWP, AP, EP, HTP, POGP	[13]
Chip	Bioethanol	1 L of 99.8% bioethanol	Thailand	GGE	[63]
Chip and wastewater	Bioethanol and Biogas	1 L of anhydrous ethanol derived from cassava	Thailand	GGE	[64]
Stems and wastewater	Biogas	1 kWh electricity and 0.09 MJ thermal energy for starch drying	Africa	GWP, AP, FEP, MEP, ECTP, MEP, HCTP, WCP	[65]
Waste closed-loop system	Biogas	1 kg cassava products (starch/flour)	Brasil	GWP, CED, FEP, AP, WDP	[42]
Roots, stems, and peel	Biogas	1 kg of cassava starch	Malaysia	CED, DF, WSI, GWP, POGP, AP, HTP, ECP	[66]
Wastewater	Biogas	1 ton of starch with 13% water content	Thailand	GGE	[67]
Wastewater	Biogas	1 MJ of bio-CNG and 1 km of vehicle driven	Thailand	GWP, HTP, AP, EP	[68]

Nomenclature: Global warming potential (GWP), Ozone layer depletion potential (OLDP), Human toxicity potential (HTP), Photochemical ozone generation potential (POGP), Acidification potential (AP), Eutrophication potential (EP), Climate change (CG), Terrestrial acidification potential (TAP), Freshwater eutrophication potential (FEP), Fossil depletion (FD), Photochemical oxidation formation (FOF), Marine ecotoxicity potential (MEP), Human carcinogenic toxicity potential (HCTP), Water consumption potential (WCP), Greenhouse gas emissions (GGE), Cumulative energy demand (CED), Deforestation (DF), Water stress index (WSI), Ecotoxicity potential (ECP), Water depletion potential (WDP).

## Data Availability

Not applicable.

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
