# Peer review of "Integrated Biorefinery and Life Cycle Assessment of Cassava Processing Residue–From Production to Sustainable Evaluation"

_plants, 2022, doi:10.3390/plants11243577_

Round 1

Reviewer 1 Report

The authors present an interesting work and discussion in the paper "Integrated biorefinery and life cycle assessment of cassava processing residue – From production to sustainable evaluation"

The article is well written and worth to publish after some minor revisions outlined below. 

1) Line 114 (2.1. Subsection should be deleted )

2) Figure 4, I suggest removing the legend and adding years on the X axis

Reviewer 2 Report

This is a well-worded overview of cassava waste management by bio-refining and an attempt to review literature data on the environmental footprint and economic consideration of such practice.

*The LCA part is way too shallow to be meaningful. LCA is the best tool we have so far to get at least a ballpark figure of the impact on the environment of products and services. The modeling is still utterly sensitive to the choice of indicators and inventory database, the availability of relevant data, and the boundaries. None of this is mentioned in the paper and, therefore, we do not get any tangible discussion or analysis of literature data in this respect. Another complication of trying to assess systems or processes that are still in the laboratory or small pilot scale is the inconsistency of data in comparison to industrial-scale processes. This is a challenge to all current efforts to analyze emerging biorefinery processes of course. My point is that all this must be discussed in the analysis of literature data to make any sense and help in interpretation. After all, this is the value of a review paper, not a long lien of citations but an analysis of the literature data in comparison. What do we learn from what is hitherto reported? The manuscript does not meet its aims here and the limited discussion provided is not meaningful. Major revision of this part will be needed.

*Also the TEA part is too shallow and the reader is left with very few insights to the viability of the different products that were made from cassava. A fewtimes, it is merely stated that "a TEA was done" with no further numbers or comments to the boundary conditions and feasibility provided.

Some additional comments for the authors' kind persual:

*Figure 1 is not fully representative of the practice of biorefineries and seems to be unbalanced toward the energy sector. Food is a well-established product from biorefineries, especially those applying algae. Materials are likewise a common target product category from biorefineries with virgin or spent biomass. Material products are generated either directly from the isolation of polymers (e.g. fibres) and from the isolation of small molecules that are fit as monomers in polymerization (do not forget the "drop-in plastics" sector). None of the above falls under the category of "biochemical".

*Line 114: define "ideal", seems ambiguous in this context.

Round 2

Reviewer 2 Report

The authors overall did a good job in addressing the comments and revising the manuscript. The manuscript reads better now, especially the LCA part which is now elaborate enough to be meaningful and add perspectives to the work done so far on Cassava biorefining. The added parts are interesting.

The TEA part is still way too shallow, actually I do not see any improvement there. It would be better to remove the part, as the readers may be misled by the abstract and aims to believe that a critical discussion will be provided (as we expect of a review paper) when there is none.

Figure 1 is not representative and can actually come off as misleading in giving an overview of biorefinery practices. The authors reply that the "intention with Figure 1 was to show in a general way the relationship and the important aspects related to raw material and the respective products in biorefineries". The point is that Figure 1 does not show that. I am sorry if my previous comment on this was not clear. Figure 1 does not accurately show the important aspects related to raw material and the respective products in biorefineries. Figure 1 is incomplete and needs revision to be publishable.
